# Reducing Administrative Burden in Clinics: Utilizing OCR and LLM Technology for Seamless External Document Handling

**Bahadır Eryılmaz[1], Helmut Becker[1],**
**Georg Lodde[3], Felix Nensa[1,2]**

**Elisabeth Livingstone[3], Dirk Schadendorf[3], Jens Kleesiek[1]**

[1]Institute of Artificial Intelligence in Medicine, University Hospital Essen
[2]Department of Radiology, University Hospital Essen
[3]Department of Dermatology, University Hospital Essen

bahadir.eryilmaz, helmut.becker@uk-essen.de
felix.nensa, jens.kleesiek@uk-essen.de
georg.lodde, elisabeth.livingstone, dirk.schadendorf @uk-essen.de

## Introduction

Integration of external clinical documents (e.g., referral letters and prior records) into electronic‑health-record workflows remains administratively burdensome. Manual scanning and data entry are costly, error-prone, and divert clinical resources from patient care. We introduce an AI-enabled workflow (Figure 1) that combines optical character recognition (OCR) with large language models (LLMs) to automatically extract and structure salient clinical information, thereby facilitating patient onboarding and treatment planning.

## Objectives

The method seeks to improve clinical throughput and reduce time spent on administrative work by delivering rapid, actionable insights from external documentation through a streamlined interface.

## Materials & Methods

The system features an intuitive interface (Figure 2) allowing clinicians to easily input documents via scanning or uploading. The OCR pipeline digitizes the text and an advanced LLM analyzes the text, performing tasks that are challenging to achieve manually in a comparable timeframe: it identifies, extracts, and structures key clinical information (e.g., diagnoses, medications, findings), generates concise summaries, and presents relevant text segments. Furthermore, clinicians can use a chat interface to obtain further information.

## Results

The application shows a clear advancement over manual document processing. It transforms unstructured external documents into structured information. This near real-time availability of key data (summaries, specific findings, medication lists) represents a profound time saving compared to manual review. Clinicians can immediately leverage this information (Figure 2), accelerating the integration of external data into patient care and decision-making far more efficiently than traditional methods allow.

## Conclusion

We present a novel, interactive application leveraging OCR and LLM technology to effectively automate the processing and integration of external clinical documents. By providing rapid insights and structured data extraction, the tool holds significant potential to improve clinician workflow efficiency, reduce administrative overhead, and enhance the timely utilization of external patient information in the clinic.

## Current Workflow

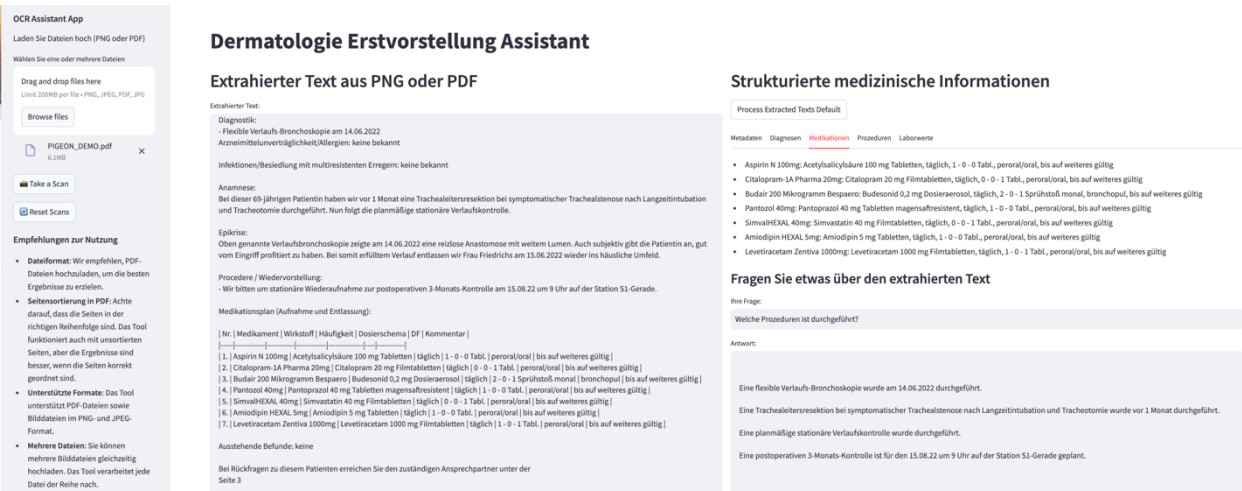

Figure 1: Proposed workflow in comparison with the current one. In the current workflow each block could take minutes but in the proposed one the steps can happen in seconds.

Figure 2: Interactive interface of the proposed automated document processing using OCR and Large Language Models. The files can be uploaded or can be scanned directly.

