# OpenReview forum: "Reducing Administrative Burden in Clinics: Utilizing OCR and LLM Technology for Seamless External Document Handling"
_NLDL.org/2026/Abstracts_Track — NLDL 2026 Abstracts_

### Official Review · Reviewer_eYPA · 2025-10-24

**Soundness:** 2
**Correctness:** 2
**Rating:** 4
**Confidence:** 4

**Summary:**

This work presents an application aimed at automating the processing of clinical documents through the use of OCR and LLM technologies. The developed interface could be of significant interest for clinical administrative workflows. However, from a methodological standpoint, the approach appears to rely directly on existing OCR and LLM tools, without clearly specifying which OCR technique or LLM model has been employed. If the authors could present a demo of how the application works at NLDL, it could be very interesting for the researchers (hence the rating).
Note: the paper is not structured according to the template suggested in the NLDL guidelines.

**Strengths:**

The work aims to enhance clinical throughput and minimize administrative time needed to process clinical documents by providing a streamlined interface.

**Weaknesses:**

An abstract section appears to be missing from the PDF. The submission was initially confusing, as the abstract in OpenReview console is formatted as a LaTeX output with several brief subsections. Including the authors' names within the abstract is unconventional and should be avoided. In summary, the paper is not structured according to the template suggested in the NLDL guidelines.

From a methodological perspective, several related approaches have already been proposed in the literature, making it unclear how the ongoing work on the OCR- and LLM-based pipeline provides specific advantages for the clinical documents analysed in this study. The focus appears to be on developing an interface for clinical use; however, both the description of the methods pipeline lack sufficient detail to assess the technical contributions. From a purely tool-development perspective, the work could be of interest if further demonstrated.

---

### Official Review · Reviewer_HHUF · 2025-10-24

**Soundness:** 3
**Correctness:** 3
**Rating:** 4
**Confidence:** 4

**Summary:**

The abstract proposes an AI-enabled workflow that combines optical character recognition (OCR) with large language models (LLMs) to automatically extract and structure information from external documents. This process can potentially reduce the adminstrative burden in clinics.

**Strengths:**

The paper proposes an innovative and potentially very useful application of LLMs in the medical field.

**Weaknesses:**

Details on how privacy/ethical concerns are addressed when the LLMs are used is missing and should be added. Are the LLMs run on-prem or is potentially personal identifiable information sent to the cloud ?

---

### Official Review · Reviewer_4sVj · 2025-10-31

**Soundness:** 3
**Correctness:** 3
**Rating:** 4
**Confidence:** 4

**Summary:**

This abstract presents an AI-enabled workflow that combines optical character recognition (OCR) with large language models (LLMs) to automatically extract and structure salient clinical information from external documents. The system aims to improve clinical workflows and reduce administrative burden by automating the processing and integration of unstructured clinical data. The proposed interface allows clinicians to upload documents, which are then analyzed by the OCR and LLM components to extract and organize key clinical information such as diagnoses, medications, and findings.

**Strengths:**

This approach addresses a challenge in clinical workflows and has the potential to reduce administrative burdens. The figure provided gives a clear overview of the proposed workflow and effectively contrasts it with the current manual workflow. This visual comparison helps to illustrate the potential improvements in efficiency and automation offered by the approach.

**Weaknesses:**

The description of the proposed workflow could be more detailed, particularly regarding the OCR and LLM components. It would be helpful to specify which OCR and LLM models are used, and to briefly explain how OCR is applied in this context. Additionally, it is unclear whether the LLM is pretrained, fine-tuned, or trained specifically for this task.
The abstract would also benefit from including short background information on the main methodological components, as well as experimental or implementation details to support the proposed workflow. A related work section is missing, and no references are provided in the introduction or methods sections.  Furthermore, a discussion of limitations should be included—particularly regarding quality assurance and privacy considerations. For example, explaining how the workflow ensures data quality, or whether privacy is maintained through local model hosting, would strengthen the credibility of the work. Finally, the conference template has not been used. We encourage the authors to address some of these points during the poster session.

---

### Decision · Program_Chairs · 2025-11-05

**Decision:**

Accept

**Comment:**

The abstract is of interest to the community and should be presented at the conference.